# A Methodology for the Multi-Point Characterization of Short-Term Temperature Fluctuations in Complex Microclimates Based on the European Standard EN 15757:2010: Application to the Archaeological Museum of L’Almoina (Valencia, Spain)

**DOI:** 10.3390/s21227754

**Published:** 2021-11-22

**Authors:** Ignacio Díaz-Arellano, Manuel Zarzo, Fernando-Juan García-Diego, Angel Perles

**Affiliations:** 1ITACA Institute, Universitat Politècnica de València, Camino de Vera s/n, 46022 Valencia, Spain; idiaz@itaca.upv.es (I.D.-A.); aperles@disca.upv.es (A.P.); 2Department of Applied Statistics, Operations Research and Quality, Universitat Politècnica de València, Camino de Vera s/n, 46022 Valencia, Spain; mazarcas@eio.upv.es; 3Department of Applied Physics (U.D. Industrial Engineering), Universitat Politècnica de València, Camino de Vera s/n, 46022 Valencia, Spain

**Keywords:** cultural heritage, microclimate monitoring, preventive conservation, EN 15757:2010

## Abstract

The monitoring and control of thermo-hygrometric indoor conditions is necessary for an adequate preservation of cultural heritage. The European standard EN 15757:2010 specifies a procedure for determining if seasonal patterns of relative humidity (RH) and temperature are adequate for the long-term preservation of hygroscopic materials on display at museums, archives, libraries or heritage buildings. This procedure is based on the characterization of the seasonal patterns and the calculation of certain control limits, so that it is possible to assess whether certain changes in the microclimate can be harmful for the preventive conservation of artworks, which would lead to the implementation of corrective actions. In order to discuss the application of this standard, 27 autonomous data-loggers were located in different points at the Archaeological Museum of l’Almoina (Valencia). The HVAC system (heating, ventilation and air conditioning) at the museum tries to reach certain homogeneous environment, which becomes a challenge because parts of the ruins are covered by a skylight that produces a greenhouse effect in summer, resulting in severe thermo-hygrometric gradients. Based on the analysis of temperatures recorded during 16 months, the air conditions in this museum are discussed according to the standard EN 15757:2010, and some corrective measures are proposed to improve the conservation conditions. Although this standard is basically intended for data recorded from a single sensor, an alternative approach proposed in this work is to find zones inside the museum with a homogeneous microclimate and to discuss next the average values collected in each area. A methodology is presented to optimize the application of this standard in places with a complex microclimate like this case, when multiple sensors are located at different positions.

## 1. Introduction

The preservation of cultural heritage for future generations is one of the main challenges facing modern societies. Regarding the preventive conservation of artworks, adequate ambient conditions are necessary to protect artifacts from potential damages in the context where these objects are kept. In this field of study, the control of environmental conditions, especially temperature (T) and relative humidity (RH), plays a key role in long-term preservation, trying to avoid irreparable damages [1,2].

In the last decades, different standards like UNI 10829 [3], DM 10/2001 [4] or The National Trust Specifications for Conservation Climate Control [5,6] have recommended values or fixed ranges of T and RH to keep artifacts in good condition. This approach based on fixed limits pretends to be universal, but it does not adapt to the changing environmental conditions with seasonal fluctuations, which are very common. If the object was in hygrostatic equilibrium under air conditions outside these fixed ranges, some of the possible consequences of artificially adapting the microclimate to follow these recommendations might be high economic costs or physical damage in objects containing organic hygroscopic materials [7].

In contrast to this fixed-range approach, the scientific community has shifted in recent decades to a more dynamic standpoint. Different methodologies have been developed starting from the analysis of historical data of ambient conditions where the object has remained for long periods of time. The target is to focus on short-term deviations with respect to those air conditions in which the object has supposedly reached an equilibrium, which are called short-term fluctuations [8,9]. This approach has led to standards such as EN 15757:2010 [10] or ASHRAE (2007) [11]. Their main objective is to promote the long-term preservation of artworks, trying to avoid physical damage like deformations or cracks in objects containing organic hygroscopic materials [12].

Different standards and guidelines have been compared in the literature. Schito [13,14] compares five performance indexes and highlights the importance of integrating HVAC systems in microclimate analysis. Holl [15] analyses the potential damages of short-term fluctuations in the indoor microclimate and compares five different guidelines from Thomson [16]; ASHRAE, class of control B and D [11]; Erhardt et al. [17]; and Mecklenburg [18]. Silva [19] emphasizes the need to collect data over long periods of time to draw strong conclusions and reiterates the importance of evaluating HVAC system performance for preventive maintenance. A complete review of preventive conservation in museum buildings has been reported by Lucchi [20].

The standard EN 15757:2010 is based on the calculation of some sort of control or safety limits, which are computed according to the variability of historical time series of RH or T with a minimum sample frequency of one record per hour. Such limits vary along the year in the case of seasonal microclimates. Firstly, the seasonal trend along the year is determined by computing a moving average. Secondly, this trajectory (i.e., sequence of values over time) is shifted up, becoming the upper safety limit, so that 7% of points appear above this limit. Likewise, the lower limit is obtained by shifting down the trajectory, so that 7% of values remain below. If the distribution of critical runs (i.e., sequence of points outside the limits) is rather uniformly distributed along the year, then RH or T conditions are considered to be adequate. This approach is intended to diagnose if thermohygrometric values are tolerable or, otherwise, if corrective actions should be adopted to reduce the risk of damage and to improve the conditions for the long-term preservation of artifacts.

Although this adaptive approach has great support from the scientific community, EN 15757:2010 establishes a general methodology that can be used in very different contexts. Hence, multiple studies have been carried out in the last years trying to clarify issues not well established by this standard, or to adapt it to specific needs [21,22,23]. This standard is usually applied for RH because it is assumed to be the most determining factor in the damage of hygroscopic materials like wood, fabric or paper. Nonetheless, several studies have also used EN 15757:2010 for the diagnosis of temperatures [21,24,25,26,27,28].

It is noteworthy that this standard does not specify how to proceed in the case of multiple sensors installed in the same room, museum, archaeological site or heritage building for the monitoring of air conditions. The recommendation would be to apply the standard to data recorded from each individual sensor, but this approach is confusing and it may lead to different conclusions even in sites keeping artworks made of the same type of materials. The installation of multiple sensors for the monitoring of indoor air conditions allows the identification and characterization of different microclimates in the context of cultural heritage, which is especially frequent in ancient churches, historic buildings or archaeological sites [26,29,30,31,32,33,34,35], some of which are equipped with HVAC systems [14,36,37]. Some studies have been carried out in museums with HVAC [38,39,40,41,42].

The approach of applying EN 15757:2010 individually to the time series from each data-logger is time consuming, and the main problem is how to reach a consensus and derive general conclusions from the set of sensors. In addition, the individual analysis from each data-logger can lead apparently to contradictory conclusions because the values appearing far out from the security limits in a certain period of time, according to one data-logger, might not correspond to the same result from the others. For example, if a given data-logger records a high temperature variability during certain time period every year, the resulting security limits will be strongly determined by such high variability, and will tend to underestimate short fluctuations that might be diagnosed as dangerous for the same type of materials in a nearby area.

The installation of multiple sensors can be useful to characterize a certain microclimate, but the application of EN 15757:2010 in this context becomes complex. Therefore, in order to address how to apply this standard in the case of multiple data-loggers, it can be assumed that short-term fluctuations affect equally at any position, regarding the long-term preservation of artifacts exhibited in a given museum, if they are composed of the same type of material. In that case, it is convenient to avoid the disparity of criteria derived from multiple sensors installed in a given site when the standard is applied separately to each data-logger.

In order to achieve this general analysis and diagnosis of the standard, the Archaeological site of l’Almoina (Valencia) is taken as a case study. This is an underground museum with the remains of the different civilizations that have existed in Valencia. Most objects on display at l’Almoina are made of stone, which is less affected by marked variations of RH than other materials like wood or paper. By contrast, sudden peaks of temperature may create thermal gradients on the surface of stone-made objects, resulting in expansion/contraction forces that might be harmful for their long-term preservation. This is the case of this site due to a skylight that allows the incidence of solar radiation, and a HVAC (heating, ventilation, and air conditioning) system which tries to keep in good condition the objects and the comfort of visitors.

In this site, 27 autonomous data-loggers were installed for the monitoring of T and RH over a period of 16 months. Past studies in this site [43,44,45] show that daily fluctuations of temperature are particularly marked in summer at certain points of the museum. Thus, it was decided to apply the proposed methodology only to T recordings in order to facilitate the interpretation of deviations from the common pattern. Other works have also applied EN 15757:2010 to the case of temperatures [21,25,26,27,28]. The present work is a contribution to the literature about indoor environments in Mediterranean climates in the context of cultural heritage [26,29,33,34,35].

In the future, RH fluctuations will be studied in order to assess the dependence between T, humidity of outside air conditions, effect of the HVAC system, and the non-negligible contributions of water diffusion by capillarity through the walls of this particular site.

The structure of this paper is as follows. Section 2 describes the materials and methods, emphasizing the data pre-processing and the description of the proposed methodologies. Next, results are presented and discussed in Section 3, making it possible to identify which zones in the museum would be convenient to pay attention to, at specific months, due to the greater short-term fluctuations recorded at those zones. Finally, conclusions and future research directions are presented in Section 4.

### Aims and Scope

The present work discusses the security limits that arise from the application of the standard in l’Almoina museum, and proposes a new methodology for the effective detection of the most sensitive areas and periods of the year. Firstly, from this case study with multiple sensors installed in the same site, the complexity of applying the standard at different points is discussed, as well as the difficulty in drawing general conclusions from the set of data-loggers. Secondly, a consensus about the security limits is discussed aimed at obtaining reasonable conclusions about the temperature variability at the different zones. Finally, a new methodology is developed in order to unify the reiterative application of the standard in the case of multiple sensors installed for microclimate monitoring. This approach provides at a glance a straightforward identification of which areas in the museum and which months of the year undergo more drastic short-term fluctuations, which is of interest to implement corrective actions in the future.

The main objectives of the present research are the following:To discuss the application of the standard EN 15757:2010 by means of a case study in the museum of l’Almoina.To explore different band widths, based on the standard, as a statistical parameter of variability aimed at discussing the microclimate in museums.To establish a new methodology for the application of the standard in the case of multi-sensor contexts aimed at easily identifying the places and periods of time with the most drastic short-term fluctuations.

## 2. Materials and Methods

### 2.1. Case Study

The Archaeological Center of l’Almoina occupies an area of about 2500 m^2^ and is located at about 3 m below the current ground level of the city. It reflects the urban history of Valencia: the Islamic Alcazar, thermal baths of the Roman Republic, the forum of the Roman Empire and other remains from the Visigoth period. Of special interest are the various monuments, the 2nd-century baths, the two streets that were the main thoroughfares in Roman times, and the courtyard of the old Muslim city. This site holds the ruins of some columns from the ancient forum, and the two main axes that organized the squared urbanism of the Roman city: *decumanus maximus*, from east to west, and the *cardus maximus*, from north to south. The point where both main streets crossed is precisely described in this archaeological site.

The main architectural feature of the museum is a transparent skylight of about 250 m^2^ that covers part of the ruins (Figure 1). It is intended to provide natural light and to allow people to see the ruins from the pedestrian square above them. Unfortunately, the incidence of solar radiation over this skylight produces marked daily variations of temperature in part of the museum underneath, particularly in summer, due to the greenhouse effect derived from the hot temperatures reached in Valencia in this season [43,44]. Being an underground museum, it is not too much affected by the outdoor conditions except for the central area under the skylight. Thus, the indoor conditions at the museum could be described as an environment with a high thermal inertia.

For the monitoring experiment, 27 data-loggers were installed at the museum to study the indoor microclimate (Figure 1). In order to record the external air conditions, an additional data-logger was located outside the museum, in a nearby building, very close to a ventilation grill. The data-logger utilized for the monitoring campaign was a MI-SOL model WS-DS102-1 [47], with a temperature range between −40 and +60 °C. According to the manufacturer, the accuracy is ±1 °C in the range of 0–50 °C and a resolution of 0.1 °C. To improve the accuracy and uncertainty of the data-logger, it was subjected to a calibration procedure described later in order to achieve compliance with EN 15758:2010.

In the museum, visitors walk through a walkway located above the ruins. The route of this walkway is indicated in Figure 1. In order to properly distribute the data-loggers inside the museum, a series of parallel and perpendicular lines, as equidistant as possible forming a grid, was established initially over the plan of this site. The purpose was to place one sensor at each line crossing. Nonetheless, the exact position of data-loggers was decided considering different additional criteria; for example, it seemed convenient to locate some data-loggers near the perimeter of the skylight, and others near the walls. In addition, the placement of sensors depends on the layout of the ruins, also trying to make them the least visible as possible to visitors. In particular, the pairs of data-loggers C0-C1 and B-D are quite close to each other. The reason was that C0 and C1 are separated by a wooden wall positioned to prevent the museum’s earthen wall from being seen by visitors. Regarding data-loggers B and D, it was decided to locate D next to an original painting that is undergoing rapid deterioration, and B could not be aligned with A due to the presence of some mosaics.

On the other hand, the number of data-loggers installed might seem somewhat excessive because the museum is unobstructed, not compartmentalized, but the purpose was to achieve an accurate characterization of the microclimate, which is of interest to identify redundant data-loggers and to decide the best location for appropriate monitoring. This issue will be tackled in a further research. The position of all data-loggers is indicated in Figure 1. Regarding the vertical position of sensors, the maximum difference of height between data-loggers is approximately 1.5 m due to the difficulty in placing them between the ruins. These differences are homogeneous throughout the museum and are considered negligible for the purposes of this study. It was assumed that thermal stratification did not occur due to the HVAC system.

Each data-logger was programmed to sample and record the T and RH every 5 min, so that a manual download of the data was necessary every two months. The monitoring experiment lasted for almost 16 months, from 10 October 2019 to 8 February 2021. The time series of temperature recorded during this time period were used for the present research study.

### 2.2. Data Pretreatment

Firstly, the records from all data-loggers were corrected, according to the bias of each sensor, and synchronized. This data pre-treatment is necessary prior to applying the methodologies proposed below in Section 2.3, Section 2.4 and Section 2.5. Subsequently, missing data were imputed by means of Principal Component Analysis (PCA), which takes advantage of the correlation structure resulting from the time series. Finally, a median filtering per hour was applied to simplify the data structure and eliminate possible anomalous observations.

#### 2.2.1. Data Calibration

As mentioned above, the accuracy of the measurements is ±1 °C according to the manufacturer. This error is excessive for the purposes of the present study, since it is of interest to characterize small differences of temperature in the same site. Thus, all sensors were calibrated prior to their installation in the museum. For this purpose, an experiment was carried out to estimate the bias of each sensor. This information was used subsequently to correct the values collected from each data-logger, in order to improve the accuracy of the recorded temperatures. The calibration experiment was performed by placing all data-loggers together during six hours inside a climatic chamber, model alpha 990-40H from Design Environmental Ltd. (Gwent, UK). Values were collected every five minutes, and three temperature levels were established: 15, 23 and 30 °C during this process. Moreover, RH was also changed at each of these levels to 75, 50 and 30% respectively over two hours. Finally, simple linear regressions were fitted for each data-logger to correct the sensor average readings according to the temperatures inside the chamber. Based on the results from this calibration experiment, the readings from each data-logger were corrected.

#### 2.2.2. Synchronization of the Time Series

A frequent problem in the comparison of time series from autonomous data-loggers is the lack of synchronization in their measurements because it is not possible to have all values collected exactly at the same instants of time. There is certain time lag between the records of the different data-loggers, which might be problematic, especially in case of sudden peaks or sharp variations. Although all devices were programmed to take readings every 5 min, one data-logger might record the values at 8:02, 8:07, 8:12, for example, while another might collect the readings at 8:04, 8:09, 8:14, and so on. These small differences are not important in case of relatively stable microclimates, but it is advisable to synchronize the readings by estimating the measurement that would correspond to the same instant of time for all data-loggers (in the example, at 8:00, 8:05 and 8:10).

In order to achieve this synchronization, a script in Python language was programmed as explained in the following example: if 20.5 °C was measured at 8:02 and 20.6 °C at 8:07, a linear interpolation is performed between both records, estimating that the reading at 8:05 would have been 20.56 °C.

#### 2.2.3. Imputation of Missing Values

Collected measurements were downloaded manually every two months by connecting the data-loggers to a laptop computer on site. In total, it was checked that 9% of the temperature values from the 27 data-loggers were missing due to the exhaustion of batteries or to human errors in the manipulation of the devices. Although 9% is a relatively low percentage, the distribution of missing data was not at random, since the time series from data-loggers A and D contained 50% missing data, followed by B and G with 23% of lost values. The lack of data is a drawback for the application of EN 15757:2010 because it does not specify how to deal with this problem. Hence, it was decided to impute all missing values.

Figure 2 shows that some sensors underwent a much higher daily variability of temperature in summer. Accordingly, in order to achieve a more precise imputation of missing data, it was decided to split the entire period under study into two terms. The first period spans from April to August, both months included, with higher daily variability recorded by sensors below or close to the skylight (Figure 2). The second period corresponds to the rest of the months.

Next, the imputation itself was carried out for each period using the Trimmed Scores Regression (TSR) imputation method, implemented in Python, which has shown good results in other works [48]. This multivariate method based on latent variables was chosen because it takes advantage of the existing correlation structure between the 27 time series. The imputation was carried out by using a PCA model built with the first 7 principal components (PCs), which explain the largest part of the overall data variability.

#### 2.2.4. Median Filtering per Hour

In order to reduce the amount of data to facilitate the application of the standard EN 15757:2010, a median filtering per hour was applied, which leads to a new time series of hourly values. This process was performed because it is a useful and simple method to avoid the presence of anomalous values that might affect the results. Furthermore, this frequency of one record per hour is convenient in this case because EN 15757:2010 can be applied on an hourly basis. A lower frequency might mask daily oscillations, which are very important for the diagnosis of short-term fluctuations, and a higher frequency is not necessary [22]. Moreover, most museums record microclimatic data with such frequency. Finally, handling large amounts of data becomes more computational time consuming.

Taking into account that one measurement was recorded every 5 min, from 10 October 2019 to 8 February 2021 (i.e., 487 days), the amount of records from each data-logger is 487 × 24 h/day × 60/5 = 140,256. By applying the median filter per hour, this quantity is reduced by 1/12.

### 2.3. PCA Clustering

PCA [49] is a very popular multivariate statistical method for studying the correlation structure of variables in large datasets. From a matrix **X** with *m* rows (observations) by *n* columns (variables), the method extracts PCs that synthesize the underlying correlation structure in the dataset. PCA is a versatile tool that can be applied for different purposes [50]. In the field of preventive conservation of cultural heritage, PCA has been used for studying the time series from multiple sensors located in the same museum, archaeological site or ancient building, allowing the characterization of similarities and dissimilarities between data-loggers, which is also of interest to group those of them with a similar performance [43,51].

According to these research works, when the time series are structured as a matrix with data-loggers in columns and time as rows, the differences between sensors are better appreciated if data are row-centered prior to applying PCA [29,43,51]. Hence, the same type of preprocessing was applied here to find groups of data-loggers with a similar pattern of temperatures, which will correspond to different microclimates existing in the museum.

The grouping of data-loggers was determined according to their projection on the multidimensional space determined by the relevant PCs and taking into account their location in the museum. Although this classification can be carried out “manually” by observing the distances between data-loggers in the multivariate space, it was decided to make this grouping automatic by using a clustering algorithm. For this purpose, the *k-means* machine-learning method [52] was applied to obtain groups of data-loggers with a similar performance.

### 2.4. Discussion of Temperature Conditions According to EN 15757:2010

In order to discuss if temperature fluctuations along the year might be potentially dangerous for preventive conservation purposes, this standard requires having data collected for at least 13 months, because a 30-day moving average is used and requires 30 more days to be calculated. The minimum frequency required is one record per hour.

Firstly, the so-called seasonally-adjusted time series is calculated from a central 30-day moving average. Secondly, the short-term fluctuations are computed by subtracting the original series to the seasonally-adjusted one. Thirdly, a range of “allowed” short fluctuations is obtained. The lower limit of this range is calculated as the 7th percentile of the short fluctuations. Likewise, the upper limit is computed as the 93rd percentile. Hence, whatever the time series, 14% of values will always appear outside these reference limits considered to be “safe” from a conservation standpoint. A series of Python scripts were programmed in order to compute the parameters according to EN 15757:2010 and to depict them graphically.

### 2.5. Comparison of Security Limits

This standard is intended to be applied for a single sensor installed for microclimatic monitoring. In the case of several data-loggers located in the same room or site, different security limits will be derived for each one, which becomes confusing. In order to properly discuss this issue, the width of these security limits (i.e., distance between the lower and upper limit) was computed according to the standard for the time series of temperature recorded by each data-logger. Next, these widths were compared and discussed, which is useful to establish a consensus criterion for the purpose of finding which data-loggers might point to areas in the museum with unfavorable conditions that might be potentially more detrimental for the preventive conservation of artworks due to excessive data variability.

An appropriate way to discuss the different time series is to decide a consensus width for the security limits to be applied for all the time series. This approach seeks to homogenize the acceptable limits for the short fluctuations in all graphs. A reasonable recommendation would be to take all band sizes and to compute the average or median as a representative. A more conservative criterion would be to choose the smaller band width, which would highlight a greater number of short fluctuations outside the security limits. Another approach would be to take the largest band size, in order to detect only those deviations assumed to be potentially most dangerous.

### 2.6. Methodology to Detect Zones and Months with Pronounced Short-Term Fluctuations

It was decided to consider the median band width, to be applied for all data-loggers. Next, it is convenient to establish a criterion to identify which months and areas in the museum require focusing the attention due to a higher presence of values outside the security limits (i.e., with an excessive variability of short-term fluctuations). In those areas it would be convenient to discuss the implementation of corrective actions.

The ideal conditions are the following. Firstly, for a given data-logger, out-of-band records should be distributed homogeneously throughout the year reflecting a random pattern, so that each month should present approximately 14% of records outside the limits. Secondly, in case of multiple data-loggers installed in the same site, short-term fluctuations should be rather similar between them. Thus, it would be reasonable to consider the same band size (i.e., width of security limits) for all data-loggers regarding the application of the standard.

In order to study whether this first ideal condition is fulfilled in this case study, a matrix of ratios was generated, containing for each data-logger (in column) and each month (in row) the value *r*_i,j_ according to Equation (1), being *out_obs*_i,j_ the number of out-of-band records in the *i*-th month, and *out_obs_TOTAL_* is the total number of out-of-band values for the *K* data-loggers during a period of *N* months.
(1)$$r_{i,j}=\frac{out\_obs_{i,j}\cdot N\cdot K}{out\_obs_{TOTAL}}\;$$

If one value is recorded every hour, the number of data per month would be approximately: 30 × 24 = 720, 14% of which should appear outside the bands, which is 100 values. In order to better understand the formula above, let us consider two data-loggers (A and B) collecting values over 3 months, which implies 600 values in total outside the limits. According to Equation (1), for the first month, *r*_1A_ = (100 × 3 × 2)/600 = 1. Thus, when this ratio is one, the percentage of data outside the limits in a certain month is equal to the average in the whole period, which is 14% according to the standard.

Once the ratios *r_i,j_* were computed, three alert levels (i.e., *safe*, *warning*, and *at risk*) were established for the values in the matrix of ratios. The safe condition was defined for *r_i,j_* ≤ 1, the maximum alert “at risk” was considered for *r_i,j_* > 2, and intermediate values were regarded as *warning*. The value 2 chosen for the maximum alert level is somewhat arbitrary and it may be argued whether this is the most appropriate value. It was determined after a preliminary study considering different values for several sets of real microclimatic data. It turned out that 2 was a reasonable criterion to easily distinguish the periods with an excessive amount of data outside the security limits. The intermediate level of alert (i.e., 1 < *r_i,j_* ≤ 2) is not based on a solid statistical foundation, but it was found to be a rather reasonable criterion according to these preliminary results.

This methodology can be applied regardless the number of months monitored, the frequency of data recording or the number of data-loggers used. Another advantage is that the method does not require a previous calibration of all sensors, because it is based on short-term fluctuations, not on accurate unbiased measurements.

## 3. Results and Discussion

For the application of EN 15757:2010, it is necessary to have at least 13 or 25 months, i.e., one or two years plus one additional month to calculate the moving average. Therefore, only the records from 16th October 2019 until 16th November 2020 were used. This period was selected for the direct application of the standard and for the methods proposed in the present work.

### 3.1. PCA Clustering

As a first step to discuss the temperatures registered by the 27 data-loggers, they were grouped based on their position in the museum and the similarity between the time series. This preliminary grouping is of interest to facilitate the comparison of results between data-loggers. Given that PCA provides a projection of the data-loggers on a low-dimensional space determined by those PCs accounting for the large amount of the data variability, it is expected that two data-loggers appearing close to each other in this space will present similar time series of temperature and, hence, the results of applying EN 15757:2010 should be similar.

For this purpose, PCA was applied after having centered the data matrix by row. Four relevant PCs were obtained; the amount of variance explained by each one (*R^2^*) was 0.51, 0.24, 0.09, and 0.05, respectively. The cumulative goodness-of-fit is R^2^ = 0.89. The loadings of these components (i.e., the weights of variables in the formation of the latent variables) allow us to establish differences between data-loggers. The matrix of these loadings was used to create groups using the *k-means* clustering method. The final grouping established 3 clusters, dividing the museum into three zones with a different microclimate: the central area under the skylight (data-loggers colored in red in Figure 1); the eastern part, corresponding to the remains of the Andalusian period (in blue), and the rest (in green).

### 3.2. Application of the Standard EN 15757:2010

By applying EN 15757:2010 to the time series recorded from a given data-logger, one graph is generated by plotting the evolution of temperature over time and depicting the seasonal trend as well as the “security” limits. This plot is illustrated in Figure 3 for the case of data-logger B2, which is located in one of the areas with the highest temperature variability. It corresponds to the most common way of visualizing the application of the standard. Another graph (Figure 4) shows just the variations with respect to the seasonal trend, which makes it easier to observe the presence of values outside the limits. The security bands are depicted in both plots, but the latter facilitates the study of short-term fluctuations.

Figure 3 shows the short-term fluctuations of temperature for B2 and highlights all values outside the security limits. This data-logger was chosen because it yields clear differences in terms of variability throughout the year depending on the season, and in daily cycles. It can be observed that most points outside the limits in B2 are concentrated in summer, between May and August. Although the methodology proposed by the standard always leads to 14% of values outside the security band, the fact that a specific period yields such a high concentration of values outside the limits can be interpreted as a risk regarding the preventive conservation of ruins in l’Almoina museum.

According to the standard, it is convenient to investigate the reason for such fluctuations in order to try to reduce them. Based on previous investigations [43,44,53], the high daily variability is explained by the skylight covering the central area of the museum. It is located above data-logger B2 and nearby ones (i.e., B3, B4, A2, A3, and C3) as shown in Figure 1. Due to the high temperatures recorded in Valencia during the summer months, this skylight generates a greenhouse effect that produces strong daily thermal fluctuations underneath. Thus, temperature varies abruptly from the early hours in the morning (minimum values reached from 2:00 to 8:00 a.m. approximately) to the hours of maximum solar radiation incident on the skylight (from 2:00 to 5:00 p.m. approximately), as shown in Figure 5.

Remarkably, in this case study, the application of EN 15757:2010 to the different time series leads to rather different results, even though all data-loggers were located in the same archaeological site. Regarding C5, it undergoes a much lower daily variability in summer (Figure 6). Unlike what happens below the skylight illustrated by B2 (Figure 3 and Figure 4), with a period of time concentrating most values outside the limits, this pattern is not observed in the case of C5 (Figure 6), given that observations outside the security band are distributed more randomly throughout the year and do not concentrate on a specific period. Moreover, the range (width) of these security limits is narrower. Therefore, short-term fluctuations recorded by C5 should receive less attention than in the case of B2 because they are potentially less aggressive or detrimental regarding the preventive conservation of the ruins.

Considering that the time series is seasonally adjusted for the application of EN 15757:2010, it is possible to appreciate more clearly the temperature fluctuations throughout the year. Two remarks can be deduced from data-loggers B2 and C5 (Figure 4 and Figure 6, respectively). Firstly, C5 also reflects a higher daily variability in summer, though values remain inside the security limits. Thus, the effect of the skylight in increasing the daily variability of temperature is also apparent in those sensors located far away from the skylight, though such effect is less intense because the incidence of solar radiation does not apply directly.

Secondly, it is very striking that the pattern of variability reflected by Figure 6 does not seem at random. Vertical dashed lines are depicted in this graph when the time series reaches a minimum. Surprisingly, in three periods, these lines are approximately equidistant, reflecting a certain regular pattern with a periodicity of approximately 3 weeks.

These regular cycles do not seem to occur in the same way throughout the year. Actually, the cycles are more attenuated in spring, between the end of February until May, and also in summer, between the end of June until mid-August. By contrast, these cycles seem to be more pronounced from mid-November to February and in June. The same pattern can also be observed for sensor B2 (Figure 4), especially during the first months. Therefore, it seems that these regularities are also apparent in data-loggers under the skylight, and might reflect a general pattern in the museum, though in the case of B2 this performance becomes less outward.

In order to further study this issue, the daily mean trajectory was computed for the average of all data-loggers inside the museum. Next, the short-term fluctuations were calculated by seasonally adjusting the time series as indicated by EN 15757:2010, as indicated in Figure 6. By using the software Statgraphics, the periodogram was computed for this time series (Figure 7), which is a powerful tool to study seasonality. The highest peaks indicate a cycle (i.e., a repetitive pattern) every 23 or 26 days. The interpretation of this result is unclear, because these periods do not correspond to a month; in case of 21 or 28 days, it might indicate certain cyclical pattern in the museum every 3 or 4 weeks, respectively, perhaps related with the function of the air conditioning system, but this is not the case.

Another hypothesis could be that cycles are determined by outdoor conditions. In order to investigate this issue, Figure 8 displays the trajectory of daily-mean temperatures recorded by the data-logger located outside the museum (in orange) together with the temperature from a nearby weather station (in blue). Both trajectories are remarkably parallel, which indicates that this data-logger provides reliable measurements. If this common pattern is compared with the daily-mean temperatures recorded by all data-loggers inside the museum (mean values, in green), certain common parallel performance is also observed: the peaks of temperature are also reflected inside the museum, though more attenuated. Likewise, the marked drops of outer temperature can be noticed in the green trajectory. It was found that the observed correlation between inner and outer temperatures did not increase if considering one or two lags in the values.

Two vertical dashed lines were plotted in Figure 8 to highlight a sudden drop of temperature that occurred inside the museum between June and September. The likely reason was that the air conditioning system was activated in the cooling mode on this period.

Results suggest that the pattern of variability throughout the year inside the museum is affected by the outer weather conditions, despite the presence of the air conditioning system, which is intended to maintain the microclimate as stable as possible. The museum is closed on Mondays, but weekly patterns that might be associated to the lack of visitors one day per week were not observed.

After the analysis of these cycles and anomalies, Figure 9 was calculated to indicate the number of out-of-band observations for each month. It can be deduced that winter months from November 2019 to January 2020 presented an excessive amount of values outside the security limits for most data-loggers, except those located below the skylight. Actually, Figure 6 shows that, in C5, the critical runs (i.e., sequences of points outside the limits) tend to be concentrated in the left part of the plot. This result is misleading because the mean temperature is lower in winter than in summer, as well as the daily variability. Hence, the fact that winter tends to concentrate a higher amount of runs outside the security limits in this particular case does not necessarily imply a risk regarding the long-term conservation of the ruins.

On the contrary, in the period from May to August 2020, the attention should be focused especially on data-loggers B2 and B3 located under the skylight, due to the high short-term fluctuations as observed in Figure 3. Curiously, September also turned out to be a month with short fluctuations to be considered by some data loggers, probably because it acts as a transition between summer and autumn.

The fact that all cells in Figure 9 appear in red for November 2019 except for B2 and B3 is explained by the higher band width for both data-loggers, due to the larger daily variability in summer. As a result, the presence of points outside the limits is concentrated in summer and not so notoriously in the other seasons. Thus, a higher daily variability in a certain season leads to security limits more separated, which might mask short-term fluctuations in other months. This consideration should be taken into account for an appropriate interpretation of results in case of microclimate monitoring experiments with multiple sensors.

### 3.3. Comparison of Band Width between Data-Loggers

The variability between data-loggers according to the band width (i.e., separation of the security limits) points to those with a higher or lower daily variability. One target for the museum is to try to reduce the temperature variability throughout the year, by getting short fluctuations as small as possible, and uniformly distributed. For this purpose, data-loggers with the highest band width will be those that should receive more attention regarding the need to implement corrective actions. Figure 10 shows the band sizes of all data-loggers, sorted according to their classification derived from PCA clustering. It can be easily distinguished that the highest band widths correspond to data-loggers below the skylight. The lowest values correspond to areas with a more stable microclimate.

In order to detect time periods and areas in the museum with excessive short fluctuations, the number of out-of-band values for each data-logger provides useful information, but this approach is time consuming and does not provide a general overview of the results. It is also convenient to study the short fluctuations using a common band width for all data-loggers, instead of considering a different band width for each one. This consensus criterion was established as the median of the set of bands, 2.50 °C. Next, out-of-band fluctuations were calculated again.

According to Figure 9, by applying the standard individually to each data-logger and computing the number of out-of-band observations, the differences between data-loggers are not clearly marked except in the case of B2 and B3. Given that the use of a different band width does not provide a clear overview of the differences, the results were recalculated after establishing a common criterion for the bandwidth (Figure 11). Results indicate that the performance of data-loggers below the skylight is quite similar to the rest from November to January. However, from April to August, the pattern is clearly distinct compared with the rest of the museum due to the effect of the skylight. Thus, differences between data-loggers are more pronounced when using a common band width. In this case, data-loggers located under the skylight (in red) present the highest number of out-of-band observations (Figure 12).

### 3.4. Methodology to Detect Zones and Time Periods with Marked Short-Term Fluctuations

According to the results, the establishment of a common criterion for all data-loggers regarding the band width is very useful to discuss the short-term temperature variability in the different zones and months of the year. However, it remains unclear which areas of the museum and periods of time should receive more attention regarding the possible risks related to the preventive conservation of artworks. For this purpose, the percentage of out-of-band observations with respect to the total was recalculated by considering the lowest band width, corresponding to data-logger F. Results are indicated in Figure 13, and cells were properly colored to highlight the highest values.

Cells colored in red in Figure 13 clearly point to those months and data-loggers where the attention should be focused on, in order to discuss if corrective actions should be necessary. Results clearly reveal that the central area of the museum under the skylight has higher short-term temperature fluctuations, particularly in summer. For the rest of the museum, most cells in red and orange correspond to December, January, June, and October. By contrast, the lowest amount of out-of-band observations correspond to March, April, July, and August.

According to Figure 9, B2 and B3 present green cells from December to March, suggesting fluctuations not excessive. However, such months are not so stable according to Figure 13 because some of these cells appear in orange. By applying this methodology, those fluctuations that according to the individualized analyses seemed less important in the central sensors like B2 according to the general analysis (Figure 9), are highlighted as a potential alert from December to February. Thus, the abrupt short-term fluctuations in summer reflected by data-loggers under the skylight do not mask the general pattern of variability, mainly characterized by somewhat regular cycles of temperatures approximately every 23 or 26 days.

The choice of the most convenient band size to be considered for all data-loggers, as already discussed, depends on how permissive we are willing to tolerate the short-term fluctuations. In this case, it seems that choosing the median size provides a clear overview about which months and zones in the museum should capture the attention of museum professionals.

Based on previous studies, it would be recommended to reduce the daily thermal variability in summer below the skylight. For this purpose, one option would be to install a canvas over the glass, or any other opaque material. This measure was tested in the year 2013 and it was effective for reducing the short-term fluctuations of temperature [44].

In summary, when multiple sensors are installed in a given site like in this case, it is recommended to carry out a preliminary data analysis by means of PCA clustering or other multivariate methods of time series analysis aimed at identifying clusters of sensors. Once identified, one approach is to discard in the future those redundant data-loggers, or to compute the average time series for each zone and apply the standard EN 15757:2010.

## 4. Conclusions

The detection of short-term fluctuations in large museums is a complex and important task for an appropriate preventive conservation. The present work addresses how to deal with temperatures recorded from multiple sensors installed for the monitoring of a complex microclimate, according to the standard EN 15757:2010. A general methodology has been established for the application of this standard in the case of multiple data-loggers located at the same site.

By monitoring the microclimate at the Archaeological Museum of l’Almoina (Valencia, Spain) as a case study, the usefulness of the standard EN 15757:2010 has been discussed in detail, and some drawbacks showed up. Drastic daily short-term fluctuations have been detected in some areas, and we discussed the effect that different periods of the year and the architecture of the museum have on them. It has also been found that an appropriate data pre-treatment is necessary to apply the EN 15757:2010 standard. To perform this in the best possible conditions, imputation of missing values is needed.

This study has also pointed out the capacity of the band ranges as a statistic capable of identifying which areas of the museum have a greater variability and, therefore, should receive more attention for an adequate preventive conservation. For this purpose, it has been useful to carry out a previous PCA-clustering.

Finally, based on the EN 15757:2010 standard, widely used in the literature, a new methodology has been established to characterize short-term temperature fluctuations in museums with complex microclimates. The advantages of this methodology are:It identifies those areas and periods of the year with a greater number of potentially dangerous short-term fluctuations. That information allows museum managers to identify the areas that require the most attention and to more easily determine what actions might be taken, for example, deciding which objects to move or which modifications to apply to the building management.It simplifies the interpretation of results through a system with three levels of alarm.Since the resulting ratio for each data-logger and month is based on the ideal conditions of homogeneity of out-of-band observations, it allows the comparison of results between different analyses from different museums.It allows more or less conservative analysis through the application of different bandwidths.

With the proposed methodology it was possible to detect those areas in the museum and time periods with marked temperature fluctuations that might be potentially problematic from a conservation standpoint. Results revealed that most of the potentially harmful short-term fluctuations correspond to the area located below the skylight, especially in summer. By contrast, data-loggers located far away from this area underwent lower fluctuations in summer. The museum of l’Almoina is very peculiar as it is a musealized archaeological site with a modern roof. The results obtained can be transferred to other similar museums or to the monitoring of new museums.

The direct application of the standard EN 15757:2010 seems insufficient to detect hidden anomalies. When multiple sensors are installed in a given site like in this case, it is recommended to carry out a preliminary data analysis by means of PCA clustering or other multivariate methods of time series analysis aimed at identifying clusters of sensors. Once identified, one approach is to discard in the future those redundant data-loggers, or to compute the average time series for each zone and apply the standard EN 15757:2010.

When the environment surrounding a given artwork suddenly changes, for example when it is moved to another museum, it is somewhat unclear how to apply the standard in order to decide if particular preventive actions are necessary to adapt the microclimate of the receiving museum to the characteristics of the previous microclimate. In this context, further research could also study the advantages of the methodology proposed here for the comparison of different microclimates.

PCA clustering turned out to be a powerful method to characterize the different microclimates in the museum. Future work should investigate how to apply other clustering techniques aimed at simplifying the application of EN 15757:2010, by establishing groups of time series with a strong similarity. In that case, it seems reasonable to apply the standard to the average trajectory.

## Figures and Tables

**Figure 1 sensors-21-07754-f001:**
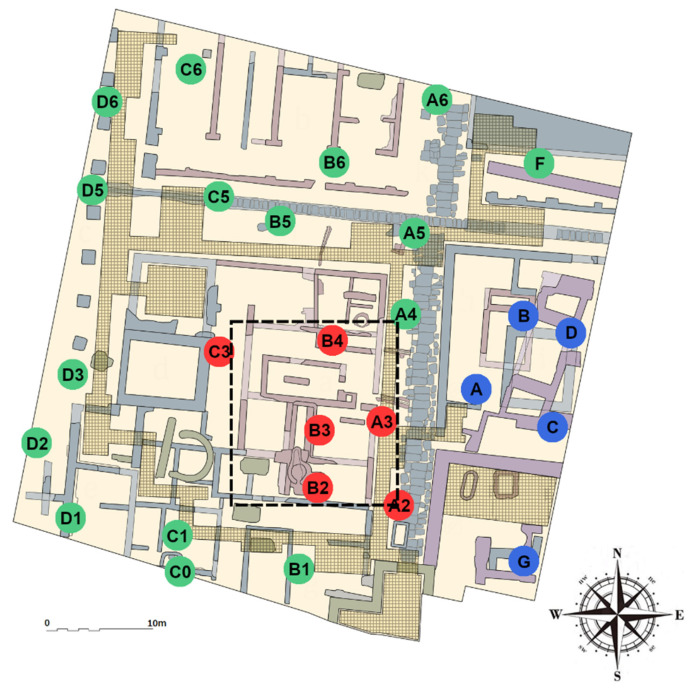
Position of 27 data-loggers installed at the Archaeological Center of l’Almoina [46]. Color codes indicate similarity in the pattern of recorded temperatures. The position of the skylight is marked as a dashed rectangle in black. The walkway used to visit the museum is indicated as a light-brown graticule (grid area).

**Figure 2 sensors-21-07754-f002:**
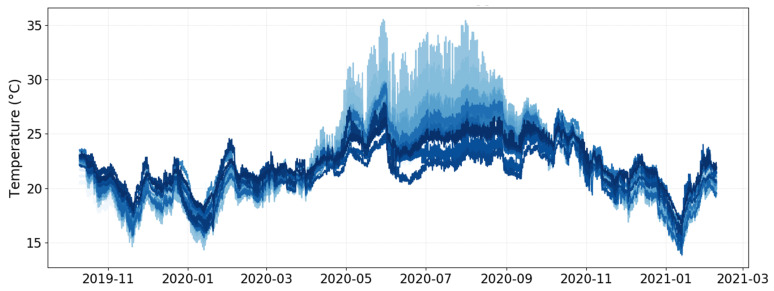
Raw trajectories of the time series of temperature recorded every 5 min by the 27 data-loggers installed in l’Almoina Museum from 10th October 2019 until 8th February 2021. Some time series contains missing values.

**Figure 3 sensors-21-07754-f003:**
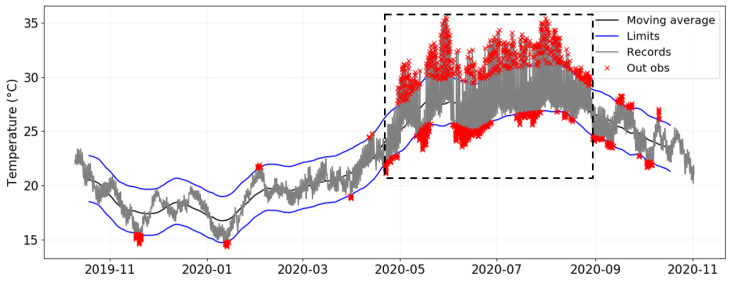
Evolution over time of temperatures recorded by data-logger B2. The seasonal trend is indicated as a black line. Security limits calculated according to EN 15757:2010 are depicted in blue. The box in dashed lines indicates the period with a higher frequency of short-term fluctuations, between 20 April and 1 September.

**Figure 4 sensors-21-07754-f004:**
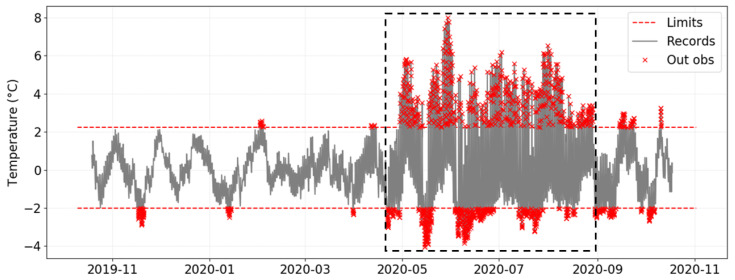
Short-term fluctuations of temperature for data-logger B2 (i.e., deviations with respect to the seasonal trend). Security limits according to EN 15757:2010 are depicted as horizontal lines in red. The box in dashed lines (same as Figure 3) indicates the period with a higher frequency of short-term fluctuations.

**Figure 5 sensors-21-07754-f005:**
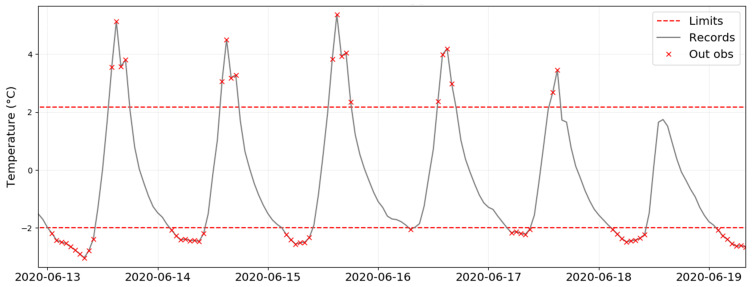
Short-term fluctuations of temperature for data-logger B2 between 13 June and 19 June (i.e., zoom of Figure 4 for those six days). It can be noticed that values outside the limits appear at specific hours every day.

**Figure 6 sensors-21-07754-f006:**
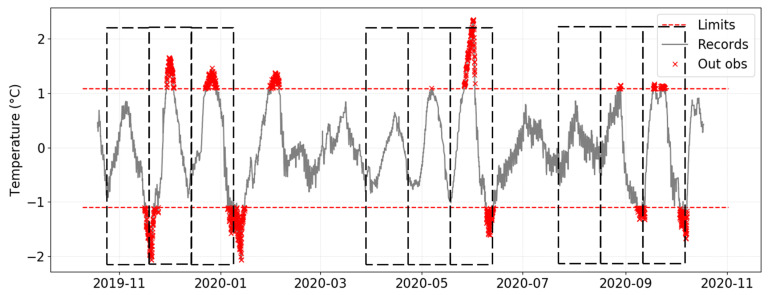
Short-term fluctuations of temperature for data-logger C5. Security limits according to EN 15757:2010 are depicted as dashed lines in red. Vertical equidistant dashed lines highlight instants of time when the time series reaches a minimum.

**Figure 7 sensors-21-07754-f007:**
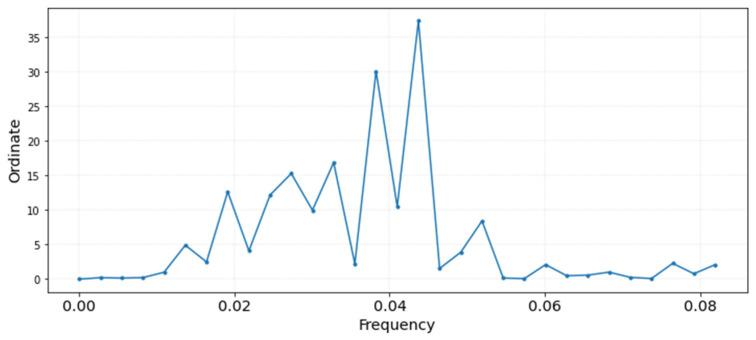
Periodogram computed with the short-term fluctuations of the mean time series of temperature (i.e., daily average of records from all data-loggers inside the museum). The two highest peaks indicate a seasonal pattern in the time series every 23 and 26 days.

**Figure 8 sensors-21-07754-f008:**
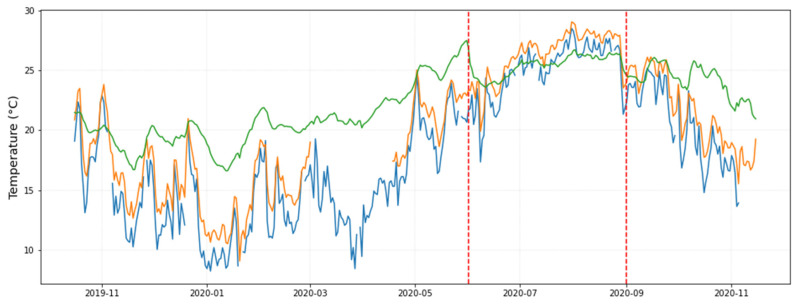
Evolution over time of daily-mean temperatures: (i) average from all sensors located inside the museum (in green), (ii) from a nearby weather station at 2 km away (in blue), and (iii) from the data-logger installed outside (in orange; part of this trajectory is missing, mainly from March to April 2020). The two vertical dashed lines indicate a period when the indoor temperature suddenly drops lower than the outdoor temperature.

**Figure 9 sensors-21-07754-f009:**
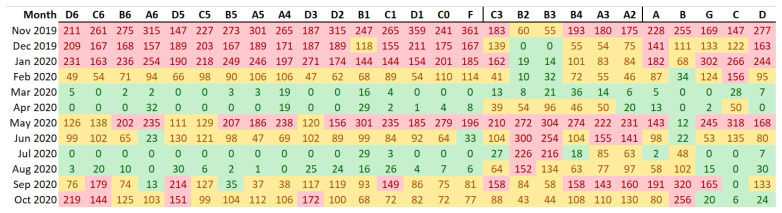
Number of observations outside the security limits for each data-logger (in columns) and each month (in rows). Such limits were determined by applying the standard EN 15757:2010 to each data-logger individually. Columns were sorted conveniently according to the three microclimates identified by PCA-clustering (i.e., the three blocks of columns correspond to the three color codes used for data-loggers in Figure 1). For each column, the highest three values are highlighted in red, and the three lowest values, in green.

**Figure 10 sensors-21-07754-f010:**
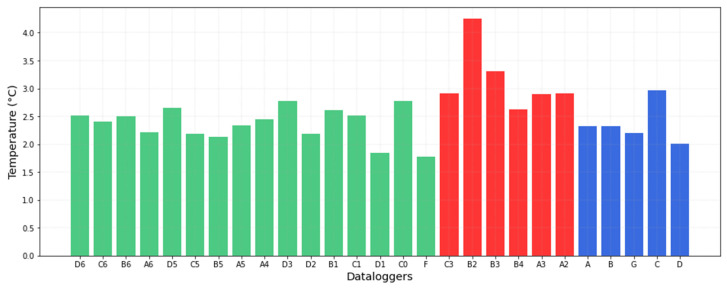
Band width (i.e., separation of security limits) for each data-logger according to EN 15757:2010. Each microclimate is represented by a different color (same as in Figure 1): the central area under the skylight in red; the ruins of the Andalusian period in blue, and the rest in green.

**Figure 11 sensors-21-07754-f011:**
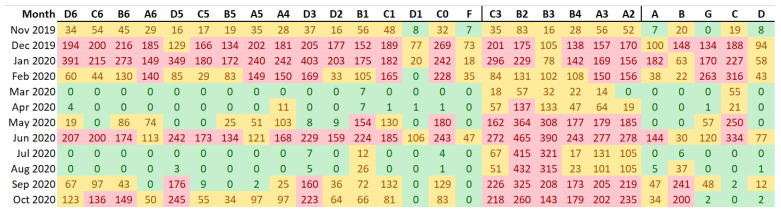
Number of out-of-band observations for each month (in row) and each data-logger (in column) by considering a common band width. Such band was computed as the median of the band resulting from applying the standard EN 15757:2010 individually to each data-logger. The lowest tercile (i.e., percentile 33) of values are highlighted in green, the highest tercile in red, and the rest in light orange.

**Figure 12 sensors-21-07754-f012:**
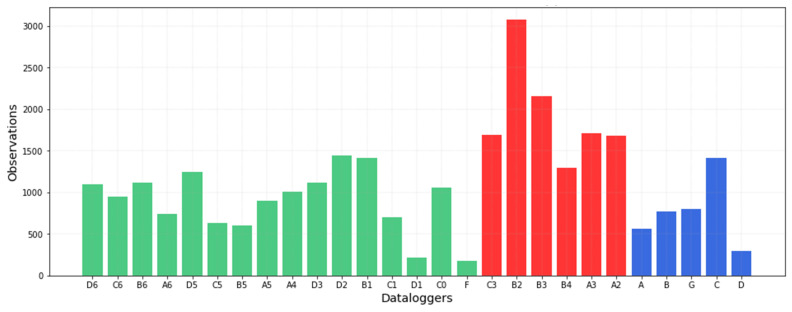
Number of out-of-band observations (i.e., values outside the security limits) for each data-logger computed by establishing the median band width (2.5 °C, see Figure 10) as a common criterion for all data-loggers.

**Figure 13 sensors-21-07754-f013:**
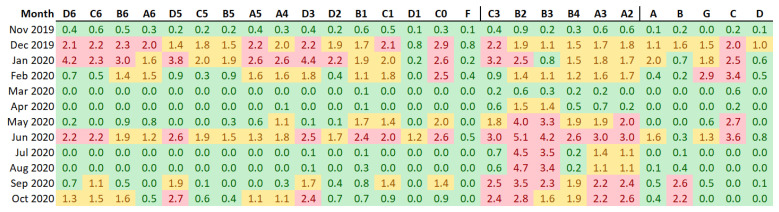
Percentage of out-of-band observations, for each month and each data-logger, with respect to the total. The number of out-of-band observations was computed by considering as a reference the median band width among data-loggers derived from EN 15757:2010. Values ≥ 2 are highlighted in red, <1 in green and the rest in orange.

## Data Availability

Datasets used are available at http://doi.org/10.5281/zenodo.4716389 (accessed on 17 November 2021).

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
