# Peer review of "A Methodology for the Multi-Point Characterization of Short-Term Temperature Fluctuations in Complex Microclimates Based on the European Standard EN 15757:2010: Application to the Archaeological Museum of L’Almoina (Valencia, Spain)"

_sensors, 2021, doi:10.3390/s21227754_

Round 1
Reviewer 1 Report
The theme is relevant and actual, aiming to characterize the climate inside church with high thermal inertia using a relevant climate monitoring campaign and using statistical tools.
The article is well organized and well written. An appropriate analysis of the state-of-the-art is made, citing several of the main research articles related to the topic. Overall, the followed methodology is robust and allows for conclusions to be drawn. The results and conclusions are supported by the measured data.
The bibliography referring to the indoor climate of cultural heritage in Mediterranean climates is still insufficient. This work explores this fragility and seeks to improve the knowledge of the behaviour of high thermal inertia buildings without air conditioning.
I have some questions I would like to see clarified:
- Is the use of hourly records adequate? I think that for churches, it would be more appropriate to record data with a lower frequency;
- Why a 7-month campaign? They do not cover the entirety of the weather seasons. A 12-month campaign would make it possible to assess the climate more rigorously and apply methods such as those described in EN 15757;
- What are the criteria to define the height of the sensors? I think some points are too close to have significance;
- I consider that the sensors should have been placed in the same vertical alignment. Why not choose this?
- I consider the use of statistical methods useful, but a little abusive for the analysis of stratification. In buildings with high thermal inertia and without air conditioning, during the summer there is a tendency for stratification. During autumn and spring, there is usually a balance, as these are transition periods. In winter, the ground will be warmer. Warm air rises, contacts the cold roof and descends, creating a convective movement. The use of statistical methods such as those used for this phenomenon can lead to biased conclusions.
Author Response
We appreciate the comments of the reviewers. All their concerns were properly addressed and taken into consideration to prepare the new revised version of the manuscript. Below you will find the list of changes explaining exactly how each of the points raised by the reviewers was addressed. As indicated by the editor, we used the “track changes” function of MS Word in order to easily track all new text that was not present in the first version and all changes introduced in the new version. Moreover, 23 new references were included in the reference list, which are highlighted in yellow.
We want to point out to readers that the lines specified in this answer may change depending on the settings used in Microsoft Word.
+The theme is relevant and actual, aiming to characterize the climate inside church with high thermal inertia using a relevant climate monitoring campaign and using statistical tools.
+The article is well organized and well written. An appropriate analysis of the state-of-the-art is made, citing several of the main research articles related to the topic. Overall, the followed methodology is robust and allows for conclusions to be drawn. The results and conclusions are supported by the measured data.
+The bibliography referring to the indoor climate of cultural heritage in Mediterranean climates is still insufficient. This work explores this fragility and seeks to improve the knowledge of the behaviour of high thermal inertia buildings without air conditioning.
"We are glad to know that the reviewer has found appropriate most parts of the manuscript.
Based on this comment of the reviewer, we have emphasized the idea of high thermal inertia (lines 194-196) for less experienced readers.
This comment of the reviewer has been introduced in the new version of the manuscript (lines 135-137), indicating a few references about buildings with high thermal inertia in Mediterranean climates."
+I have some questions I would like to see clarified:
- Is the use of hourly records adequate? I think that for churches, it would be more appropriate to record data with a lower frequency;
Based on this question raised by the reviewer, this discussion has been implemented in the new version of the manuscript (lines 317-320) with references that clarify the question."
+ Why a 7-month campaign? They do not cover the entirety of the weather seasons. A 12-month campaign would make it possible to assess the climate more rigorously and apply methods such as those described in EN 15757;
"The experiment of microclimate monitoring started on October 10th 2019 and finished on February 8th 2021, as indicated in (lines 238, 308, 322 and 427). In total, there are 16 months available with measurements. Probably the reviewer has misunderstood this issue."
+ What are the criteria to define the height of the sensors?
"Based on this comment of the reviewer, a few lines were introduced in the new version of the manuscript (lines 231-235)."
+I think some points are too close to have significance;
"We agree with the reviewer that some sensors appear to be too close to each other in Figure 1, like C0 vs. C1, and B vs. D. This issue, regarding the procedure to decide the best location for the data-loggers, is explained in lines 218-220. The reason for such close position of these sensors is the following: data-loggers C0 and C1 are located to opposed sites of a wooden wall installed to avoid that the earth wall of the museum is seen (this is an excavated museum). By contrast, data-loggers B and D were positioned as indicated in Figure 1 for the following reasons: D was located next to an original painting that is undergoing rapid deterioration, and B could not be aligned with A due to the presence of some mosaics."
+ I consider that the sensors should have been placed in the same vertical alignment. Why not choose this?
"We agree with the reviewer that the most convenient option would be to place the data-loggers with the same vertical alignment. However, this criterion was not possible in this case because the placement of sensors depends on the layout of the ruins and where the museum management allowed us. In addition, a requirement of the museum was to install the sensors so that the devices were as less visible as possible to visitors. This comment has been introduced in the new version of the manuscript (lines: 201-102, 212-213 and 218-230)."
+ I consider the use of statistical methods useful, but a little abusive for the analysis of stratification. In buildings with high thermal inertia and without air conditioning, during the summer there is a tendency for stratification. During autumn and spring, there is usually a balance, as these are transition periods. In winter, the ground will be warmer. Warm air rises, contacts the cold roof and descends, creating a convective movement. The use of statistical methods such as those used for this phenomenon can lead to biased conclusions.
"As indicated by the reviewer, in places with a marked thermal inertia there might be a clear stratification. However, in this case, due to the air conditioning system, the air is mixed at a reasonable speed, and we considered that the effect of air stratification was not relevant. For this reason, data-loggers were positioned to zero elevation, with a maximum elevation difference of 1.5 m (this issue has been added in lines 231-235). Most parts of the museum are at a similar elevation, about 3.5 meters from the roof, so we considered that the effect of air stratification may not be in this case as interesting as in other historical buildings."
Reviewer 2 Report
The paper presents a methodology for the multi-point characterization of short-term temperature fluctuations in museum buildings.
The introduction is focused on standards and guidelines. You consider all the standards on this topic, but not all the books and the guidelines. Several institutions published guidelines for defining correct values of temperature relative humidity in museums, but it is better to focus on standards. Probably, the reference to a review of preventive conservation in museum buildings can help you on this part. In the introduction lacks a part on the gap in literature, to show the novelty of your methodology. Several Papers have been in this topic have been published by Camuffo, Schito, Lucchi, Kramer in museum buildings. For example the paper https://doi.org/10.3390/su12125155 proposed a multi-points monitoring in a museum building, for discover bing the thermal fluctuations. It just one paper, but several paper gave been published on this. Your paper is different, as it is more technical in the analysis if data. Show this part both in the introduction inserting similar paper in the aims, and in the conclusion as a comparison. Several references are in churches or in historic buildings. It is better to focus on museums, as several
Papers have been published in this buildings. To show better the aims is necessary to introduce a specific section on it. Some date in the case study are inserted both in the introduction and in the case study description. Avoid any repetition. Improve the conclusion showing the main outcomes and the transferability of your study in others research.
Author Response
We appreciate the comments of the reviewers. All their concerns were properly addressed and taken into consideration to prepare the new revised version of the manuscript. Below you will find the list of changes explaining exactly how each of the points raised by the reviewers was addressed. As indicated by the editor, we used the “track changes” function of MS Word in order to easily track all new text that was not present in the first version and all changes introduced in the new version. Moreover, 23 new references were included in the reference list, which are highlighted in yellow.
We want to point out to readers that the lines specified in this answer may change depending on the settings used in Microsoft Word.
Based on these considerations of the reviewer, significant changes have been introduced in the introduction, research design and conclusions. These modifications are explained in the comments of the reviewer.
+The paper presents a methodology for the multi-point characterization of short-term temperature fluctuations in museum buildings. The introduction is focused on standards and guidelines. You consider all the standards on this topic, but not all the books and the guidelines. Several institutions published guidelines for defining correct values of temperature relative humidity in museums, but it is better to focus on standards. Probably, the reference to a review of preventive conservation in museum buildings can help you on this part.
"Based on this comment of the reviewer, we have added a new paragraph (lines 60-69). In the same paragraph, different comparisons of standards and guidelines have been referenced. The following reference to a review about preventive conservation in museum buildings has been inserted: Lucchi, E. Review of preventive conservation in museum buildings (lines 67-68)."
+In the introduction lacks a part on the gap in literature, to show the novelty of your methodology. Several Papers in this topic have been published by Camuffo, Schito, Lucchi, Kramer in museum buildings. For example the paper https://doi.org/10.3390/su12125155 proposed a multi-points monitoring in a museum building, for discovering the thermal fluctuations. It just one paper, but several paper have been published on this.
"Based on this comment of the reviewer, the paper mentioned has been cited (line 99) as well as other studies published by Camuffo, Schito, Lucchi, Kramer and other authors (lines 98-99). The novelty of the study has been specified from the line 161."
+Your paper is different, as it is more technical in the analysis of data. Show this part both in the introduction inserting similar paper in the aims, and in the conclusion as a comparison. Several references are in churches or in historic buildings. It is better to focus on museums, as several papers have been published in these buildings.
"Based on this comment of the reviewer, 5 new references to preventive conservation in museums have been added (line 99)."
+To show better the aims is necessary to introduce a specific section on it.
"A new specific section (1.1. Aims and scope) has been added starting on the line 148."
+Some date in the case study are inserted both in the introduction and in the case study description. Avoid any repetition.
"Based on this comment of the reviewer, the reiterations in the introduction have been integrated into the case study section."
+Improve the conclusion showing the main outcomes and the transferability of your study in others research.
"Based on this comment of the reviewer, the conclusions have been improved by emphasizing the various contributions of the study, especially the proposed methodology (lines 709-741)."
Round 2
Reviewer 2 Report
Thanks for considering my suggestions